# Comparison of a Nanofiber-Reinforced Composite with Different Types of Composite Resins

**DOI:** 10.3390/polym15173628

**Published:** 2023-09-01

**Authors:** Zümrüt Ceren Özduman, Burcu Oglakci, Derya Merve Halacoglu Bagis, Binnur Aydogan Temel, Evrim Eliguzeloglu Dalkilic

**Affiliations:** 1Department of Restorative Dentistry, Faculty of Dentistry, Bezmialem Vakif University, 34093 Istanbul, Turkey or zozduman@bezmialem.edu.tr (Z.C.Ö.);; 2Private Clinic, Çekmeköy, 34794 Istanbul, Turkey; mrvhal@yahoo.com; 3Department of Pharmaceutical Chemistry, Faculty of Pharmacy, Bezmialem Vakif University, 34093 Istanbul, Turkey

**Keywords:** composite, fiber, polymer, nano, mechanical, physical properties

## Abstract

The aim of this study was a comprehensive evaluation and comparison of the physical and mechanical properties of a newly developed nano-sized hydroxyapatite fiber-reinforced composite with other fiber-reinforced and particle-filled composites. Commercially available eight composite resins (3 fiber-reinforced and 5 particle-filled) were used: Fiber-reinforced composites: (1) NovaPro Fill (Nanova): newly developed nano-sized hydroxyapatite fiber-reinforced composite (nHAFC-NF); (2) Alert (Pentron): micrometer-scale glass fiber-reinforced composite (µmGFC-AL); (3) Ever X Posterior (GC Corp): millimeter-scale glass fiber-reinforced composite (mmGFC-EX); Particle-filled composites: (4) SDR Plus (Dentsply) low-viscosity bulk-fill (LVBF-SDR); (5) Estelite Bulk Fill (Tokuyama Corp.) low-viscosity bulk-fill (LVBF-EBF); (6) Filtek Bulk Fill Flow (3M ESPE) low-viscosity bulk-fill (LVBF-FBFF); (7) Filtek Bulk Fill (3M ESPE) high-viscosity bulk-fill (HVBF-FBF); and (8) Filtek Z250 (3M ESPE): microhybrid composite (µH-FZ). For Vickers microhardness, cylindrical-shaped specimens (diameter: 4 mm, height: 2 mm) were fabricated (n = 10). For the three-point bending test, bar-shaped (2 × 2 × 25 mm) specimens were fabricated (n = 10). Flexural strength and modulus elasticity were calculated. AcuVol, a video image device, was used for volumetric polymerization shrinkage (VPS) evaluations (n = 6). The polymerization degree of conversion (DC) was measured on the top and bottom surfaces with Fourier Transform Near-Infrared Spectroscopy (FTIR; n = 5). The data were statistically analyzed using one-way ANOVA, Tukey HSD, Welsch ANOVA, and Games–Howell tests (*p* < 0.05). Pearson coefficient correlation was used to determine the linear correlation. Group µH-FZ displayed the highest microhardness, flexural strength, and modulus elasticity, while Group HVBF-FBF exhibited significantly lower VPS than other composites. When comparing the fiber-reinforced composites, Group mmGFC-EX showed significantly higher microhardness, flexural strength, modulus elasticity, and lower VPS than Group nHAFC-NF but similar DC. A strong correlation was determined between microhardness, VPS and inorganic filler by wt% and vol% (r = 0.572–0.877). Fiber type and length could affect the physical and mechanical properties of fibers containing composite resins.

## 1. Introduction

Composite resins find extensive application as restorative materials in contemporary dentistry. Clinical lifespan of composite resin restorations can vary due to factors related to the tooth, risks associated with the patient, chosen restorative techniques, and the materials implemented [1]. Adverse outcomes, such as marginal mismatch, marginal discoloration, microleakage, recurrent caries, and postoperative sensitivity, encountered in composite resin restorations are typically attributed to polymerization shrinkage. Studies have found that the volumetric shrinkage of composite resins ranges from 1% to 6% [2]. Another common problem is fractures in composite resins applied to deep, large cavities. When subjected to stress, these fractures can be attributed to various mechanical properties of the composite resin, including fracture strength, elasticity, and marginal degradation. These factors can be assessed through tests, such as flexural properties and fracture strength [3].

The filler amount and types, organic monomers (e.g., Bis-GMA and TEGDMA) inorganic and organic matrix interfaces can affect the mechanical and physical properties [4,5,6]. The mechanical characteristics of composites can be improved to withstand chewing stresses by modifying the filler particle size and morphology. This modification has been shown to improve their overall performance. Regarding the type of filler used, composites can be categorized into fiber-reinforced and particle-filled composites [7,8]. Fibers are crucial for reinforcing composites by acting primarily as crack stoppers, thus enhancing mechanical properties that closely resemble those of natural tissues. The effectiveness of fiber–reinforced composite resins is significantly influenced by microstructural properties, including fiber loading, fiber orientation, fiber diameter, fiber length, and the adhesion between fibers and the polymer matrix. These composite resins strengthen teeth and mimic dentin’s stress-absorbing characteristics, making them suitable for direct restoration of extensive cavities in vital and devital posterior teeth [9]. 

Several manufacturers have produced short fiber–reinforced composites (SFRCs) to address the limitations of conventional composite resins. These SFRCs attempt to structurally resemble the fibrous composition of dentin, and some are specifically recommended for bulk bases in high-stress-bearing areas. According to fiber aspect ratio theory, these materials can be classified as either high aspect ratio SFRCs with short fiber lengths on a millimeter scale or low aspect ratio SFRCs with short fiber lengths on a micrometer scale. One of the commercial forms of high aspect ratio SFRC is EverX Posterior (GC Corp., Tokyo, Japan), which consists of polyethylene and glass fibers, in millimeters (mm) scale. Low aspect ratio SFRC is Alert (Pentron, Wallingford, CT, USA), which is a dimethacrylate-based material with glass fibers in micrometer (µm) scale [10]. A new fiber-containing composite launched in 2018 has been added to composites reinforced with calcium-phosphate (hydroxyapatite). NovaPro Fill (Nanova, MO, USA) is noteworthy for being the first composite to incorporate fiber filler material with dimensions that are measured in nanometers, representing an innovation on the nano-scale [11].

Microhybrid composites are frequently preferred as particle-filled composite resins in the posterior region due to high wear resistance. Recently launched bulk-fill composites also belong to the particle-filled composite class, and many brands produce them using different technologies. Different types of bulk-fill composite resins can be classified according to their viscosity: flowable (low-viscosity) and sculptable (high-viscosity) bulk-fill composites. The viscosity of bulk-fill composites is influenced by factors such as the type and quantity of fillers used in the composite as well as the content of the organic matrix. Bulk-fill composites have been launched on the market with claims of polymerizing in thicker layers (4–5 mm), showing lower polymerization shrinkage, and possessing physical properties that improved upon conventional composite resins [12,13]. 

Limited data can be found in the literature comparing the properties of newly developed hydroxyapatite nanofiber-reinforced composites to different types of composite resins. Thus, in this study, a total of 8 commercial composite materials were utilized, categorized into 2 distinct composite groups: particle-filled and fiber-reinforced. The particle-filled composites were further divided into 2 subgroups: bulk-fill and microhybrid composites. As for the fiber-reinforced composites employed in this research, they were classified into 3 subgroups: nanometer, micrometer, and millimeter categories. The aim of the present study was a comprehensive evaluation and comparison of the physical and mechanical properties of a newly developed nano-sized hydroxyapatite fiber-reinforced composite with other fiber-reinforced composites (micrometer- and millimeter-scale fiber-reinforced) and particle-filled composites (low-viscosity and high-viscosity bulk-fill and microhybrid composites) used in the posterior region. 

The null hypothesis of this in vitro study was as follows: 

There would be no differences in microhardness, flexural strength, modulus elasticity, volumetric polymerization shrinkage, or degree of conversion between nanofiber-reinforced composite and other fiber-reinforced or particle-filled composites. 

## 2. Materials and Methods

### 2.1. Sample Size Calculation

G * power 3.1 program was used for the purpose of sample size calculation based on the estimated effect size between groups. In this study, for each group, a minimum of 5 specimens were necessary to obtain a medium effect size (d = 0.50, 80% power, 5% type 1 error rate).

The composite resins used in this study, group names and their compositions are shown in Table 1.

A single operator prepared specimens according to the manufacturer’s instructions during all analyses. 

### 2.2. Microhardness Measurements

Cylindrical-shaped specimens (height: 2 mm, diameter: 4 mm) of each composite resin were prepared using Teflon molds (n = 10) (Figure 1). The specimens were sandwiched between two transparent mylar strips and glass slides to achieve a smooth, polymerized surface. The excess material was then removed by applying pressure using the glass slides. Subsequently, the specimens were subjected to light polymerization through the upper glass slide with a light-emitting diode light-curing unit (LED LCU; Valo, Ultradent, UT, USA, irradiance of 1000 mW/cm^2^) for 20s. Periodically during specimen fabrication, a radiometer (Bluephase Meter II, Ivoclar Vivadent, Schaan, Liechtenstein) was used to control the light intensity to ensure it remained at that level. A permanent marker was used to mark the bottom surfaces. The specimens were then placed in a dark vial containing distilled water at 37 °C for 24 h. A Vickers microhardness test was performed with an HMV Microhardness Tester (HMV-G, Shimadzu Corp., Kyoto, Japan) according to the ASTM E384-17 standard [14]. Three measurements were obtained on each sample’s top and bottom surfaces (200 g load, 10 s dwell time). Vickers hardness values were recorded as the average of these measurements. The hardness value of the bottom surface was divided by the hardness value of the top surface to determine the hardness ratios (%), and these ratios were later converted into percentages.

### 2.3. Flexural Strength and Modulus Elasticity Measurements

ISO 4049 guidelines were followed to prepare 10 bar-shaped specimens (2 × 2 × 25 mm) for each composite resin (Figure 2). A half-split stainless steel mold was utilized in the preparation process [15]. A single increment of composite resin was placed into the mold, and it was then covered on both sides using two transparent mylar strips, along with glass slides.

A universal testing machine (AGS-X, Shimadzu Corp., Kyoto, Japan) was used for the three-point bending test at a 1.0 mm/min crosshead speed until fracture (Figure 3). The span length for the test was set at 15 mm. Analysis software (Trapezium X, Shimadzu Corp., Kyoto, Japan) was used to record the load-deflection curves and calculate the flexural strength (FS) using Equation (1) [16]:FS = 3F/L(2wt^2^)(1)

Within this equation, “F” represents the maximum force, “L” denotes the distance between the supports, “w” signifies the width, and “t” indicates the thickness of the sample. The modulus elasticity (E) was determined by calculating the slope of the linear region in the load-deflection curve. This calculation was executed by dividing the load (F) by the displacement (d) in the linear elastic region. Equation (2) was used for this calculation.
E = (F/d)(L^3^/[4wt^3^])(2)

### 2.4. Volumetric Polymerization Shrinkage (VPS) Measurements by AcuVol Video Image Analysis

VPS was measured using a video image device (AcuVol, Bisco, Schaumburg, IL, USA) at 25 °C in single-view mode. Low-viscosity composites (LVBF-SDR, LVBF-EBF, and LVBF-FBFF) were syringed onto a 4.2-mm-diameter polytetrafluoroethylene (PTFE) pedestal, high-viscosity composites (µH-FZ, nHAFC-NF, mmGFC-EX, µmGFC-AL, and HVBF-FBF) were rolled into a ball and placed on the PTFE pedestal in front of a CCD camera (n = 6). A resting period of five minutes was granted to the specimen to mitigate the impact of slumping on the measurement. The light-curing tip was fixed 1 mm above the top of the specimen and was then polymerized for 20 s with a LED LCU (1000 mW/cm^2^). The volumetric shrinkage was measured five minutes after light polymerization to allow the specimen’s temperature to stabilize at room temperature. Using the images taken before and after polymerization, the VPS rate was measured with this formulation and recorded as a percentage:(V1–V2)/V1 × 100(3)

Within this equation, “V1” represents the volume of the specimen before light polymerization; “V2” represents the volume of the specimen after light polymerization.

### 2.5. Degree of Conversion (DC) Measurements 

Five specimens were prepared using cylindrical molds (height: 2 mm, diameter: 5 mm; n = 5) to determine the polymerization degree of conversion (DC) measurements. Each composite resin was placed into the molds and covered on both sides with two transparent mylar strips and glass slides to remove excess material and avoid oxygen inhibition. The light-curing tip was fixed on the top glass slide and the specimens were polymerized with an LED LCU (1000 mW/cm^2^) for 20s. Periodically during specimen fabrication, a radiometer (Bluephase Meter II, Ivoclar Vivadent, Schaan, Liechtenstein) was used to control the light intensity to ensure it remained at that level. After light curing, they were kept at 100% humidity at 37 °C for 24 h in a dark vial [11].

After polymerization, the degree of conversion (%) was calculated for the top and bottom surfaces with an attenuated total reflectance Fourier transform infrared spectrometer (ATR-FTIR; ALPHA Bruker spectrometer with a platinum-ATR accessory). Each composite resin was also measured by the ATR-FTIR before polymerization (Figure 4).

The spectra were obtained using the following parameters to calculate the DC: a wave number range of 4000–600 cm^−1^, 16 scans per spectrum, and a spectral resolution of 4 cm^−1^. The stretching vibrations of the aliphatic C = C bonds at 1636 cm^−1^ served as the analytical absorption band, while the aromatic C = C bonds at 1607 cm^−1^ served as the internal reference absorption band. The DC value was determined by calculating the ratio of the peak heights of the analytical and reference absorption bands, which were then normalized by the ratio of the uncured monomers, as shown in the following equation: (4)$$DC\%=1-\left[\frac{\left(\frac{A_{1}}{A_{2}}\right)polymer}{\left(\frac{A_{1}}{A_{2}}\right)monomer}\right]\times 100$$

This calculation is represented by the equation above, where A_1_ and A_2_ represent the peak intensities of the aliphatic C = C (1636 cm^−1^) and aromatic C = C (1607 cm^−1^) bonds, respectively. The subscripts outside the parentheses refer to the spectra before (monomer) and after (polymer) light curing. The mean top and bottom DCs and standard deviations were determined for each type of material.

### 2.6. Statistical Analysis

A software program (SPSS 22.0 Windows, SPSS Inc., Chicago, IL, USA) was used. The Shapiro–Wilk and Levene’s tests determined the variables’ normality and the variances’ homogeneity for all data. Since the data were normally distributed but had heterogeneous variances for microhardness data, a Welsch’s ANOVA test was used to compare the materials. All pairwise comparisons were performed with the Games–Howell test. Since the data were normally distributed and had homogeneous variances for flexural strength, modulus elasticity, VPS, and DC values, a one-way analysis of variance (ANOVA) test was used to compare the materials. All pairwise comparisons were performed using the Tukey HSD test. Pearson coefficient correlation was performed to determine the linear correlation between microhardness, flexural strength, VPS and inorganic filler content. A significance level of 0.05 was considered for all analyses.

## 3. Results

### 3.1. Microhardness Measurements

The mean microhardness values (HV) and standard deviations of all groups are shown in Table 2. 

On the top surfaces, compared to other groups, Group µH-FZ showed the statistically highest HV. Comparing the fiber-reinforced composites, Group mmGFC-EX and Group µmGFC-AL showed statistically higher microhardness than Group nHAFC-NF. Comparing the particle-filled composites, Group HVBF-FBF showed statistically higher HV than Group LVBF-FBFF, Group LVBF-SDR, and Group LVBF-EBF. 

On the bottom surfaces, compared to other groups, Group µH-FZ had the statistically highest HV. Comparing the fiber-reinforced composites, Group mmGFC-EX exhibited statistically higher HV than Group µmGFC-AL and Group nHAFC-NF. Comparing the particle-filled composites, Group HVBF-FBF exhibited statistically higher HV than Group LVBF-FBFF, Group LVBF-SDR, and Group LVBF-EBF. 

Group nHAFC-NF (63.21%) and Group µmGFC-AL (62.65%) exhibited a lower hardness ratio than threshold values (80%). Other tested groups showed a hardness ratio exceeding the 80% threshold values.

### 3.2. Flexural Strength and Modulus Elasticity Measurements

The mean flexural strength (MPa), modulus elasticity (GPa) values, and standard deviations (±SD) of all the tested groups are shown in Table 3.

Compared to other groups, Group µH-FZ and Group mmGFC-EX exhibited the statistically highest flexural strength and modulus elasticity. Comparing the fiber-reinforced composites, Group nHAFC-NF showed similar flexural strength and modulus elasticity to Group µmGFC-AL. Comparing the particle-filled composites, Group HVBF-FBF exhibited similar flexural strength and modulus elasticity to Group LVBF-FBFF, Group LVBF-SDR, and Group LVBF-EBF.

### 3.3. VPS Measurement by AcuVol Video Image Analysis

The mean VPS (%) values and standard deviations (±SD) with the AcuVol video image analyzer for all groups are shown in Table 3. 

Compared to other groups, Group HVBF-FBF showed the statistically lowest VPS. Comparing the fiber-reinforced composites, Group mmGFC-EX had a lower VPS than Group nHAFC-NF. Comparing the particle-filled composites, Group µH-FZ had statistically lower VPS than Group LVBF-EBF, Group LVBF-FBFF, and Group LVBF-SDR. 

### 3.4. DC Measurements

The DC values (%) and standard deviations (±SD) of all groups at the top and bottom surfaces are shown in Table 4. 

At the top and bottom, compared to other groups, Group LVBF-SDR had the lowest DC, and Group nHAFC-NF had the highest DC.

At the top, comparing the fiber-reinforced composites, Group mmGFC-EX showed statistically higher DC than Group µmGFC-AL, while it showed a similar DC to Group nHAFC-NF. Comparing the particle-filled composites, Group µmH-FZ exhibited a similar DC to Group LVBF-SDR, Group HVBF-FBF, Group LVBF-FBFF, and Group LVBF-EBF. 

At the bottom, comparing the fiber-reinforced composites, Group mmGFC-EX had a similar DC to Group nHAFC-NF and Group µGFC-AL, and Group nHAFC-NF had a statistically higher DC than Group µGFC-AL. Comparing the particle-filled composites, Group µH-FZ had a similar DC to Group LVBF-SDR, HVBF-FBF, LVBF-FBFF, and LVBF-EBF. 

### 3.5. Linear Correlation between Physical and Mechanical Properties and Inorganic Filler Content 

Linear correlation analysis for all groups comparing microhardness, flexural strength, VPS and filler content (wt/vol%) is shown in Table 5. A very strong correlation (r = 0.841–0.877) on the top and a moderate correlation (r = 0.572–0.668) on the bottom, were detected between microhardness and inorganic filler by wt% and vol%. A weak correlation (r = 0.364–0.385) was found between flexural strength and inorganic filler by wt% and vol%. A strong correlation (r = 0.776–0.860) was present between VPS and inorganic filler by wt% and vol%.

## 4. Discussion

Despite significant advancements in restorative composites, fiber-containing restorative materials still exhibit two primary limitations: inadequate mechanical strength and polymerization shrinkage. Several studies on nanofiber-reinforced composites have focused on the orientation and distribution of fibers. There has been growing interest in enhancing composites by incorporating nanofibers. Within the currently available literature, published studies concerning restorative composites reinforced with nanofibers are limited. In this research, comprehensive evaluation and comparison of the physical and mechanical properties of a newly developed nano-sized hydroxyapatite fiber-reinforced composite with other fiber-reinforced composites (micrometer- and millimeter-scale fiber-reinforced), and particle-filled (low-viscosity and high-viscosity bulk-fill and microhybrid) composites used in the posterior region, was performed. Based on the findings, the null hypothesis that there would be no differences in microhardness, flexural strength, modulus elasticity, volumetric polymerization shrinkage, or degree of conversion between nanofiber-reinforced composite and other fiber-reinforced or particle-filled composites, was rejected.

The surface hardness of composite resins in posterior stress-bearing areas is a crucial mechanical property influenced by the effectiveness of polymerization and bonding between monomers [17]. Various factors related to resin composites, such as the size, shape, and fraction of fillers in the inorganic phase, can influence hardness. Hardness typically increases with higher filler content. The specific composition and structure of the organic matrix also impact hardness [18]. The properties of dental composites are associated with the volume fraction of fillers incorporated within the resin and the effectiveness of the silanization procedure used to connect the filler and matrix phases. Stress within the material is primarily transferred through interactions between hard particles [19]. 

In this study, the microhybrid composite (82% wt) showed significantly higher microhardness on the top and bottom surfaces than the other composites. High-viscosity bulk-fill (76.5% wt), also showed significantly higher microhardness than low-viscosity bulk-fill composites. These findings align with several studies [19,20,21] of higher filler loading. Pearson coefficient correlation confirmed the fact that a positive correlation was determined between microhardness and filler content. Moreover, when comparing the fiber-containing composites, millimeter-scale fiber-reinforced composites exhibited significantly higher microhardness than nano-sized hydroxyapatite fiber-reinforced composites. The millimeter-scale glass fiber-reinforced composite contained fibers with lengths ranging from 1 to 2 mm, surpassing the critical fiber length. Consequently, incorporating these short fibers into a resin matrix could significantly improve mechanical properties. Uyar et al. [22] showed that aligning nanofibers enhanced dental composites’ mechanical properties. Thus, the lack of alignment of nanofibers may have contributed to lower microhardness values for nanofiber-reinforced composites. Hardness values of bottom/top surfaces generally can be used to measure the degree of polymerization. In the literature, it was indicated that an acceptable degree of polymerization is considered successful if the bottom hardness corresponds to a minimum of 80% of the hardness of the top surface [23]. However, in this study, a hardness ratio lower than 80% was found for nanofiber-reinforced composite and micrometer-scale glass fiber-reinforced composite. 

Flexural strength measures a material’s ability to withstand maximum stress, such as chewing loads before it fails. It functions as an indicator of the durability of a restorative material when subjected to stress. Modulus elasticity describes the stiffness of the material [24,25]. These flexural properties can be influenced by factors such as filler size, morphology, and the amount of filler in the restorative material. Increased filler content and smaller spherical-shaped fillers typically increase packing density and enhance mechanical properties. However, other notable factors, including stress transfer between the matrix and the filler particles and adhesion between these components, can also impact flexural strength [26]. 

In the present study, microhybrid and millimeter-scale glass fiber-reinforced composites demonstrated significantly higher flexural strength and modulus elasticity than the other composites. This finding is in line with Yancey et al. [11] who reported that nanofiber-reinforced composite showed significantly lower flexural strength and modulus elasticity than microhybrid composite. Lassila et al. indicated that nanofiber-reinforced composite showed significantly lower modulus elasticity than millimeter-scale glass fiber-reinforced composite [9]. Millimeter-scale glass fiber-reinforced composite, with its 76% wt filler load and inclusion of glass fibers inside the polymer matrix, exhibited superior flexural properties [10]. This finding aligns with previous research by Suzaki et al., who also noted the excellent flexural characteristics of millimeter-scale glass fiber-reinforced composite [27]. In the present study, nanofiber-reinforced composite displayed a flexural strength and modulus elasticity similar to micrometer-scale glass fiber-reinforced composite. The nanofiber-reinforced composite contains fibers with diameters in the nanometer scale (50–200 nm) and lengths ranging from 100 to 150 μm, while micrometer-scale glass fiber-reinforced composite has fibers with diameters of 7 μm whose length is in the micrometer scale (20–60 μm). Those fibers containing materials have properties that fall below the critical fiber length and the desired aspect ratio (the ratio between length and diameter) [28]. Researchers have also noted that specific fiber-to-polymer ratios and resin concentrations can cause a decrease in flexural properties. This decrease has been attributed to limitations in the bonding between the fibers and the resin matrix or incomplete resin infiltration, leading to voids that compromise the material’s strength. Among the many parameters reinforcing the fibers’ efficiency, the fibers’ orientation is especially important [29].

To assess the VPS of composite resins, AcuVol was chosen as the preferred method for its ease of use and cost-effectiveness. VPS is considered an unfavorable side effect of composite resins [30]. The composition of composite resin, including the types of organic monomers (e.g., Bis-GMA and TEGDMA) and the types and content of inorganic fillers, can impact VPS. TEGDMA, with its lower molecular weight and higher number of double bonds, tends to increase VPS more than Bis-GMA [31]. However, an increase in filler content or the presence of pre-polymerized filler particles can result in decreased VPS. These factors influence the extent of VPS in composite resins. According to the literature, VPS values for high-viscosity resin composites typically range from 1% to 3%, while low-viscosity composites may exhibit VPS values of up to 6% [32]. In the current study, the VPS of all tested composites fell within the range of approximately 1% to 5%, which is considered clinically acceptable. Notably, the high-viscosity bulk-fill composite demonstrated a significantly lower VPS than the other tested materials. High-viscosity bulk-fill composite resin includes UDMA (urethane dimethacrylate) and two innovative methacrylate monomers: aromatic urethane dimethacrylate (AUDMA) and addition–fragmentation monomer (AFM). The purpose of incorporating these monomers is to specifically target and minimize the VPS in the composite material. The mechanism of action of AUDMA and UDMA, a high molecular weight monomer, reduces the number of reactive groups. This reduction helps moderate the volumetric shrinkage and stiffness of the polymer matrix, which in turn contributes to the development of polymerization stress. By minimizing the reactivity of the monomers, AUDMA and UDMA can effectively mitigate the extent of volumetric shrinkage and the associated stress during polymerization [33]. The presence of AFM in high-viscosity bulk-fill composite can reduce polymerization shrinkage stress without negatively affecting the polymer’s physical properties. A combination of AUDMA, UDMA, and AFM reduces polymerization shrinkage stress while maintaining the favorable physical properties of the polymer composite [34]. This finding could be attributed to this restorative material’s AFM and AUDMA monomer. 

In this study, the microhybrid composite (82% wt) showed significantly lower VPS than low-viscosity bulk-fill composites (LVBF-SDR 70.5% wt, LVBF-Filtek Bulk Fill Flow/64.5% wt, and LVBF-Estelite Bulk Fill/70% wt). The lower VPS values may be attributed to the higher filler composition of the microhybrid composite. Besides, the Pearson coefficient correlation indicated a strong correlation between VPS and inorganic filler content. The literature has indicated that fiber–reinforced restorative materials can also function as a stress-distributing, energy-absorbing mechanism for reducing VPS while strengthening mechanical properties [35,36,37]. Comparing the fiber-containing composites, millimeter-scale glass fiber-reinforced composite exhibited significantly lower VPS than nanofiber-reinforced composite. 

The DC usually represents the degree of polymerization or the percentage of polymerizable double bonds converted to a single bond. The ideal composite resin material should have lower VPS and higher DC properties. Extensive documentation supports the notion that the high molecular weight Bis-GMA molecule, possessing a rigid aromatic backbone and robust intermolecular hydrogen-bonding capability, exhibits limited molecular mobility, reducing conversion within the polymer network [38]. Introducing the low-viscosity ethoxylated counterpart, known as Bis-EMA, as a partial replacement for Bis-GMA can enhance the crosslinking monomers’ molecular reactivity [39]. Furthermore, the incorporation of low-molecular-weight TEGDMA and UDMA, recognized for their heightened mobility, contributes to an overall increase in the ultimate conversion process. 

In this study, at the top and bottom, nanofiber-reinforced composite showed a similar DC to millimeter-scale glass fiber-reinforced composite while it exhibited significantly higher DC than micrometer-scale glass fiber-reinforced composite. The nanofiber-reinforced composite contains Bis-EMA, UDMA, TEGDMA, and no Bis-GMA, while millimeter-scale glass fiber-reinforced composite material consists of Bis-GMA, randomly distributed millimeter-scale length glass fibers and filler particles. The components of this material are expected to impede the penetration of activating light. However, in this study, one possible explanation for the observed higher DC values in millimeter-scale glass fiber-reinforced composite is the utilization of two photoinitiators, namely camphorquinone and a derivative of trimethylbenzoylphosphine oxide. This combination produces more free radicals than camphorquinone alone, thus enhancing this material’s DC [39]. The particle size of inorganic fillers can also significantly affect light scattering and, subsequently, DC properties. Xu et al. determined that the DC values of nanofiller resin composites were higher than those of microfiller resin composites [2]. Therefore, the nanofiber-reinforced composite’s higher DC values could be explained by its filler particle size. Besides, the finding of this study is in line with Yancey et al., who reported that nanofiber-reinforced composite showed higher DC at the top and bottom than microhybrid composite [12]. 

## 5. Conclusions

In conclusion, A strong correlation was determined between microhardness, VPS and inorganic filler by wt% and vol%. Fiber type and length could affect fiber-containing composite resins’ physical and mechanical properties. Microhybrid composite displayed the highest microhardness, flexural strength, and modulus elasticity, while high-viscosity bulk-fill exhibited significantly lower VPS than other composites. When comparing the fiber-containing composites, millimeter-scale glass fiber-reinforced composite showed significantly higher microhardness, flexural strength, modulus elasticity, and lower VPS than nanofiber-reinforced composite but similar DC.

## Figures and Tables

**Figure 1 polymers-15-03628-f001:**
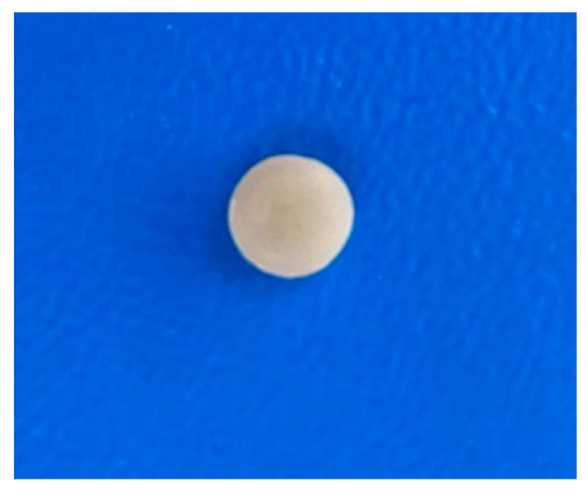
A specimen prepared for microhardness test.

**Figure 2 polymers-15-03628-f002:**
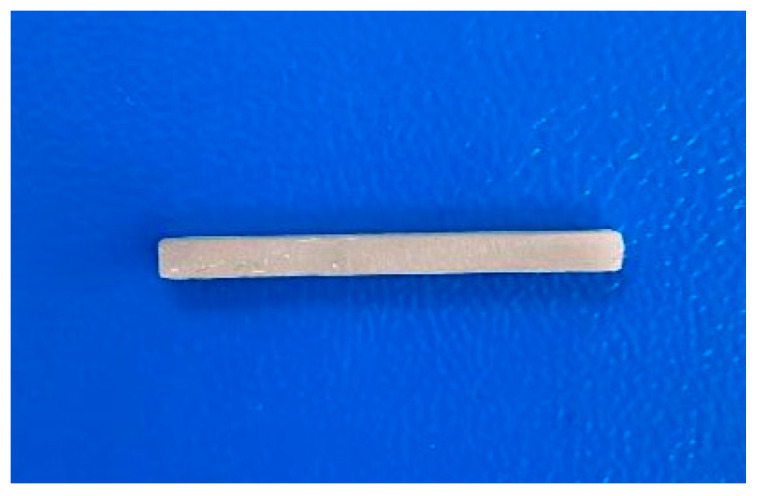
A specimen prepared for three-point bending test.

**Figure 3 polymers-15-03628-f003:**
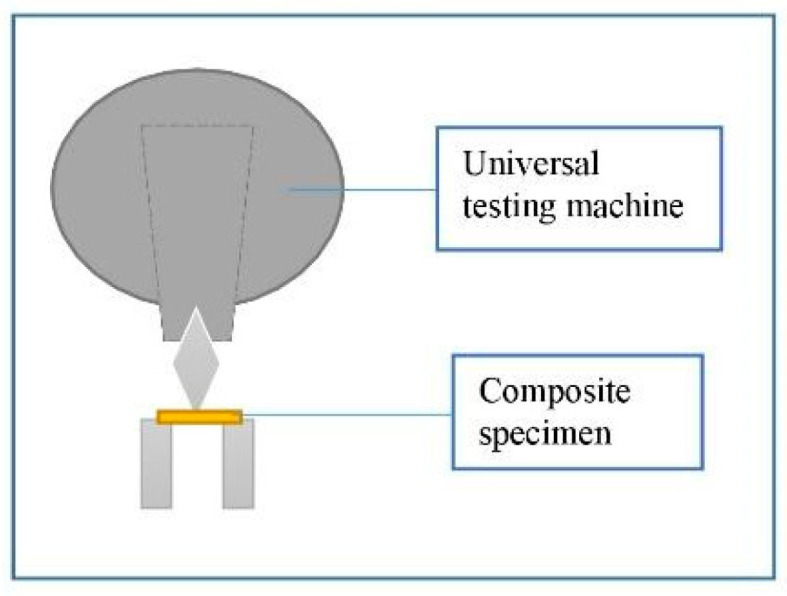
Schematic representation of three-point bending test.

**Figure 4 polymers-15-03628-f004:**
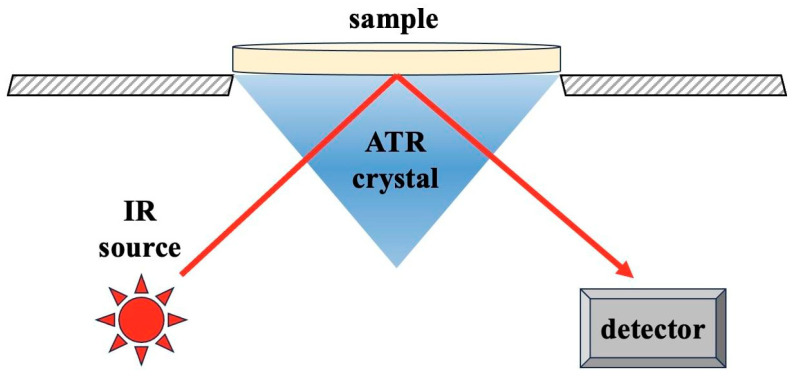
Schematic representation of attenuated total reflectance Fourier transform infrared spectrometer (ATR-FTIR) system.

**Table 1 polymers-15-03628-t001:** Composite resins used in this study and their compositions.

	Clasification	Brand Name	Manufacturer	Filler Particle	Filler %	Organic Matrix
Fiber-reinforced composites	Nano-sized hydroxyapatite fiber-reinforced Composite	NovaPro Fill (nHAFC-NF) (A2 Shade) Lot No.: 050035	(Nanova, MO, USA)	Barium silicate glass, amorphous fumed silica, hydroxyapatite fiber(diameter in nanometer scale (50–200 nm) and length in range between 100 and 150 μm)	77 wt%, NA	Bis-EMA, UDMA, TEGDMA
	Micrometer-Scale Glass Fiber-Reinforced Composite	Alert (µmGFC-AL) (A2 Shade) Lot: 7015315	(Pentron, CT, USA)	Silica and micrometer scale length glass fiber (20–60 μm) and diameter of 6–10 μm	84 wt%, 62 vol%	Bis-GMA, UDMA, TEGDMA, THFMA
	Millimeter-Scale Glass Fiber-Reinforced Composite	EverX Posterior (mmGFC-EX) (Universal Shade) Lot: 1904183	(GC Corp., Tokyo, Japan)	Millimetre scale length glass fiber filler, Barium glass	76 wt%, 57 vol%	Bis-GMA, PMMA, TEGDMA
Particle-filled composites	Low-Viscosity Bulk-Fill Composite	SDR Plus Bulk Fill (LVBF-SDR) (A2 Shade) Lot: 2/2306000334	(Dentsply, DE, USA)	Silanated barium-alumino-fluoro-borosilicate glass; silanated strontium alumino-fluoro-silicate glass; surface treated fume silicas; ytterbium fluoride; synthetic inorganic iron oxide pigments, and titanium dioxide.	70.5 wt%/47.4 vol%	Modified UDMA, TEGDMA, dimethacrylate, trimethacrylate resins
	Low-Viscosity Bulk-Fill Composite	Estelite Bulk Fill (LVBF-EBF) (A2 Shade) Lot: E699	(Tokuyama Corp., Tokyo, Japan)	Supranano spherical filler Silica, Zirconia, Ytterbium trifluoride Filler particle size 200 nm	70 wt%/56 vol%	Bis-GMA, Bis-MPEPP, TEGDMA,
	High-Viscosity Bulk-Fill Composite	Filtek Bulk Fill (HVBF-FBF) (A2 Shade) Lot: N979068	(3 M ESPE, MN, USA)	Nonaggregated silica filler (20 nm), nonaggregated zirconia filler (4–11 nm), aggregated zirconia/silica cluster filler (20 nm silica/4–11 nm zirconia), and an agglomerate ytterbium trifluoride filler (100 nm)	76.5 wt%/58.4 vol%	AFM, AUDMA, UDMA, DDDMA
	Low-Viscosity Bulk-Fill Composite	Filtek Bulk Fill Flow (LVBF-FBFF) (A2 Shade) Lot: N934595	(3 M ESPE, MN, USA)	The zirconia/silica particles [size range of 0.01–3.5 µm (The average particle size is 0.6 µm)] Ytterbium trifluoride (particle-size range of 0.1–5.0 µm)	64.5 wt%/42.5 vol%	TEGDMA, BisGMA, Bis-EMA, Procrylat and UDMA
	Microhybrid-filled composite	Filtek Z250 (Z250) (µH-FZ) (A2 Shade) Lot: N968746	(3 M ESPE, MN, USA)	Silica/Zirconia, cluster fillers Filler particle size of 0.01–3.5 μm (average 0.6 μm)	82 wt%, 60 vol%	TEDGMA, UDMA, Bis-EMA

Abbreviations: Bis-EMA, ethoxylated bisphenol A dimethacrylate; UDMA, urethane dimethacrylate; TEGDMA: triethylene glycol dimethacrylate; Bis-GMA, bisphenol A-diglycidyl dimethacrylate; THFMA, tetrahydrofurfuryl methacrylate; PMMA, Polimetil metakrilat; Bis-MEPP, bisphenol A ethoxylate dimethacrylate; AFN, addition fragmentation monomer; AUDMA, aromatic dimethacrylate; DDDMA, 1,12-dodecane dimethacrylate; Bis-EMA, ethoxylated bisphenol A glycol dimethacrylate, vol, volume; wt, weight.

**Table 2 polymers-15-03628-t002:** Mean microhardness values (HV) and standard deviations (±SD) of top and bottom surfaces for all groups; bottom/top microhardness ratio (%).

Groups	Top (HV)	Bottom (HV)	Bottom/Top (%)
nHAFC-NF	62.15 ± 2.33 ^c^	39.29 ± 7.85 ^ab^	63.21
µmGFC-AL	72.67 ± 3.39 ^d^	45.53 ± 5.50 ^b^	62.65
mmGFC-EX	71.43 ± 2.74 ^d^	68.53 ± 0.94 ^d^	95.94
LVBF-SDR	37.04 ± 2.27 ^a^	33.22 ± 2.67 ^a^	88.82
LVBF-EBF	48.73 ± 1.39 ^b^	47.23 ± 2.56 ^b^	96.92
HVBF-FBF	69.08 ± 3.22 ^d^	61.30 ± 3.44 ^c^	88.73
LVBF-FBFF	35.73 ± 1.00 ^a^	34.17 ± 1.10 ^a^	95.63
µH-FZ	96.54 ± 2.07 ^e^	87.12 ± 2.48 ^e^	90.24
* **p** *	**<0.001**	**<0.001**	

Different lower-case letters indicate the significant differences within the given column (*p* < 0.05).

**Table 3 polymers-15-03628-t003:** The mean flexural strength, modulus elasticity and VPS values and standard deviations (±SD) for all groups.

Groups	Flexural Strength (MPa)	Modulus Elasticity (GPa)	VPS (%)
	Mean ± SD	Mean ± SD	Mean ± SD
nHAFC-NF	134.32 ± 25.09 ^ab^	12.27 ± 2.29 ^ab^	3.33 ± 0.19 ^cd^
µmGFC-AL	107.66 ± 51.58 ^a^	9.83 ± 4.71 ^a^	2.76 ± 0.22 ^bc^
mmGFC-EX	226.25 ± 37.19 ^c^	20.66 ± 3.40 ^c^	2.62 ± 0.34 ^b^
LVBF-SDR	140.61 ± 26.13 ^ab^	12.84 ± 2.39 ^ab^	4.12 ± 0.46 ^ef^
LVBF-EBF	138.09 ± 23.66 ^ab^	12.61 ± 2.16 ^ab^	3.73 ± 0.45 ^de^
HVBF-FBF	162.05 ± 38.28 ^b^	14.80 ± 3.50 ^b^	1.89 ± 0.39 ^a^
LVBF-FBFF	117.16 ± 36.93 ^ab^	10.70 ± 3.37 ^ab^	4.71 ± 0.47 ^fg^
µH-FZ	259.26 ± 42.99 ^c^	23.68 ± 3.93 ^c^	2.72 ± 0.12 ^bc^
** *p* **	**<0.001**	**<0.001**	**<0.001**

Different lower-case letters indicate the significant differences within the given column (*p* < 0.05).

**Table 4 polymers-15-03628-t004:** The mean DC values (%) and standard deviations (±SD) of all groups at the top and bottom surfaces.

Groups	Top	Bottom
nHAFC-NF	60.56 ± 2.96 ^e^	52.44 ± 4.52 ^c^
µmGFC-AL	48.18 ± 2.88 ^ab^	43.26 ± 2.17 ^ab^
mmGFC-EX	56.18 ± 2.32 ^de^	49.56 ± 4.01 ^bc^
LVBF-SDR	46.66 ± 3.38 ^a^	40.22 ± 3.34 ^a^
LVBF-EBF	50.40 ± 1.61 ^abc^	45.38 ± 3.38 ^abc^
HVBF-FBF	55.00 ± 4.05 ^cde^	48.32 ± 2.28 ^bc^
LVBF-FBFF	53.62 ± 2.11 ^bcd^	47.32 ± 4.96 ^abc^
µH-FZ	51.44 ± 1.75 ^abcd^	46.20 ± 1.81 ^abc^
** *p* **	**<0.001**	**<0.001**

Different lower-case letters indicate the significant differences within the given column (*p* < 0.05).

**Table 5 polymers-15-03628-t005:** Linear correlation analysis between microhardness, flexural strength, VPS and inorganic filler content (wt/vol%).

	Correlation Coefficient (r)	*p*
**Filler %wt-microhardness bottom**	**0.572**	**0.139**
**Filler %wt-microhardness top**	**0.877**	**0.004**
**Filler %wt-flexural strength**	**0.364**	**0.375**
**Filler %wt-VPS**	**−0.776**	**0.023**
**Filler %vol-microhardness bottom**	**0.668**	**0.101**
**Filler %vol-microhardness top**	**0.841**	**0.018**
**Filler %vol-flexural strength**	**0.385**	**0.394**
**Filler %vol-VPS**	**−0.86**	**0.013**

## Data Availability

Not applicable.

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
