# Peer review of "Comparison of a Nanofiber-Reinforced Composite with Different Types of Composite Resins"

_polymers, 2023, doi:10.3390/polym15173628_

Round 1

Reviewer 1 Report

The manuscript describes the comparison of the physical and mechanical properties of a newly developed nanofiber-reinforced composite with seven different composite resins used in dentistry.

The title states it was the “comparison of a Nanofiber-Reinforced Composite with Different Types of Composite Resins”, which means 1 vs 7? However, the results and discussion were not presented in line with this 1 vs 7 rational.

Although authors state “Fiber type, length, volume fraction, aspect ratio, orientation, and interfacial adhesion could significantly affect …properties…” the description of each parameter for each sample type is very vague, leading to weak and not well supported conclusions.

Besides this, result presentation needs to be improved, for example, Tables 3-5 can be combined into one, Line103-111 is repetitive of Table 1. Presentation is after all boring, with no graph but only tables.

English usage needs to be improved, as well as some inappropriate usage, such as “Line 50 …based on…”“vital and non-vital…” both disk-shaped and cylindrical have been used for the similar sample geometry, etc.

“Since the data were normally distributed for microhardness data, a Welsch’s ANOVA test was used to compare the materials. All pairwise comparisons were performed with the Games–Howell test. Since the data were normally distributed for flexural strength, modulus elasticity, VPS, and DC values, a one-way analysis of variance (ANOVA) test was used to compare the materials.” It appears the 2 sets of data were both normally distributed but were treated differently.

needs to be improved.

Reviewer 2 Report

Dear Authors,

Thank you for your submission. Please consider the following comments:

Abstract

Line 23: ‘This study compared the physical and mechanical properties of a newly developed nano-sized glass fiber-reinforced composite’ Compared with what?

Introduction

Which was the criterion for choosing the selected materials?

Please examine more carefully your in-text citations which in many cases are irrelevant and not focusing in what you want to support. For example, in line 46 ‘Composites typically have an average lifespan of 5.7 years’. I could not find any such statement in the paper of Sakaguchi et al., and, in any case, the paper is outdated and does not examine the lifespan of composites, but their polymerization contraction.

Also, in line 97: ‘A power analysis was performed to establish the sample size according to the litera-97 ture (12, 13)’. Again, the citations are not relevant. You should have a statistical analysis article instead.

Line 84: “NovaPro Fill (Nanova, C lumbia, MO, USA) is the first composite containing nano-sized fiber filler’ (intro) give the definition of nano sized. Also, give the definition of microhybrid (line 90).

Materials and Methods

Table 1: please reorganize the Table so that the reader is shown the synthesis of the materials, in particular the critical/relevant features, such as concentration of fibers, their chemical identity and dimensions. Also include the coding of your groups, removing it from the list above.

Please include photos of specimens and procedures, so that the reader can understand and follow the steps of your experiments.

Please include more details of the experimental procedure and its critical attributes: Shade of composites, calibration and testing of the irradiation, time of irradiation, orientation and distance of the light-polymerization unit etc.

How did you decide about the storage time and conditions for your specimens (distilled water at 37°C for 24 h for microhardness, and flexural strength, lines 129, 141, and dry at 37°C for 24 h in a dark vial for degree of conversion, line 166) For the volumetric change, the specimens were measured immediately after, or kept for 24 h as the rest of the specimens?. What effect would the storage time and environment on the polymerization processes occurring in the specimens and thus on the property values you found?

Results:

Some graphs illustrating the measured values and differences would be helpful for the reader to visualize the findings.

All the Tables in the Results section do not include the units of measurement of the properties. Please add them.

Table 2: You did not analyze the difference in microhardness between top and bottom surface, for the same sample. These differences are indirectly indicative of the depth of cure. The depth of cure is considered adequate when the difference between top and bottom surface is higher than 80%. Thus you can reach some conclusions about the depth of cure.

Table 6: Similarly, the difference of degree conversion between top and bottom of the same material would be of great interest.

It would be advisable to add a linear correlation analysis, connecting the filler content of the materials with their scores in microhardness flexural strength and polymerization shrinkage.

Discussion

In the beginning of the discussion, I did not see any comparison with published data from existing literature, any more than the statement that this information is scarce (line 248). There are however, such publications. For example, the work of Yancey et al. ‘Properties of a New Nanofiber Restorative Composite. Oper Dent. 2019;44(1):34–41 (reference No 10). They examined many of the materials you used, with the same purpose as yours, with the aim to evaluate the NovoPro Fill material, and tested the same properties, i.e. flexural strength/modulus, degree of conversion, depth of cure, and polymerization shrinkage. Thus I would expect to see a comparison with these and other published findings in your discussion.

Conclusions

Rephrase the conclusions in order to point out what is the answer to your research question, after the experimental procedure instead of replicate the findings.

The English language and style of the manuscript are adequate for ensuring readability and clarity. I did not detect any serious grammar mistakes or inappropriate sentence structure.

Reviewer 3 Report

1. Should write the authors' affiliation following the format of the journal's template. 2. In the analysis of the sample calculation, could you provide a little more explanation of the information on the effect size and the final sample sizes? It does not make much sense that when the same parameters are used for the different parts of the study, each part has a different size. 3. Could you please add in Table 1 the code of each group and the type of composite described in the above list? In addition, it would also be good if you could add the batch number information for each material studied. 4. In the description of microhardness it is not necessary to add (n=10) as you start the sentence by saying that 10 samples were prepared per material; there is no need to repeat information. 5. Change the word "samples" to "specimens". The term "sample" is more associated with the statistical concept of sample size rather than the specimens used in a test. Please, check the text and change every time it is needed. 6. What are the dimensions of the specimens used in the VPS test? 7. Just as you have described what the letters mean in equations 1 and 2, you should describe equation 3, so that the difference between V1 and V2 is clear. 8. You should standardise the format numbers in the text. In English is usually always use dots, for example 62,15 replaced by 62.15. Check and change in the whole text. 9. In reference 26 the name of the magazine is missing, please check that the same thing has not happened in any other reference.

Round 2

Reviewer 1 Report

After the first round of revision, the overall quality has been improved.

However, the major shortcoming still persists, due to the mixed material classifications: “nano-sized glass fiber-reinforced composite and different types of composites (SFRC composites, low/ high-viscosity bulk-fill composites and microhybrid composite).” Consistant classification criterion is key.

The result presentation and discussion need to be modified, because currently the MS has been written for speciallised dentistry journals, comparing commercial products to inform clinical practice. However, to qualify for publicaiton in a multi-disciplinary journal Polymers, this needs to be modifed, because in the broader dental materials and polymer science context each individual commercial name does not convey meaningful information.

The null hypothesis has not been revisited. Also this should be changed to reflect the main findings of the study, the correlation.  Perhaps the title should also be changed.

Thus, a major revision is recommended.

Details:

Please use consistant classificaiton, based on glass fiber types (also since the focus is primarily “on fiber type”): roughly nHA-glass, glass, glass+zirconia, mixed glasses, etc. And if the authors prefer (and secondary focus is on “length parameters”) further classification could be made based on length scale: mm, um, nm, etc. Please read the literature to find appropriate classifications.

Table 1 should be accordingly modified, which can definitely use more columns to increase the readability, e.g. for classification, manufacturer, filler types, filler %. Take out word“group” througout.

In Abstract the introduction of the 8 commercial samples needs to be consistent because right now some have the manufactuers, and some do not. And it is not clear which one is the “newly developed nano-sized glass fiber-reinforced composite.” Two styles of decimal point have been used. Please only use one.

Please use proper multiply sign rather than letter x.

The highly impacting work depicting the mechanical properties of restorative dental composites in general, factors influencing the properties, filler-matrix interface and filler reinforcement effect should be cited doi:10.1038/ncomms9631

Author Response

Dear Editor,

We would like to say thank you to the reviewers. We sincerely appreciate all valuable comments and suggestions, which helped us to improve the quality of the article. Our responses to the Reviewers’ comments are described below in a point-to-point manner. We have addressed all the comments in paragraphs as explained below. It should be noted that our revised parts in the manuscript has been highlighted as well.

After the first round of revision, the overall quality has been improved.

Response: Thanks for the reviewer for helpful feedback. Thank you for taking the time to review the revised version of our manuscript. Your guidance and input have been invaluable in enhancing the clarity and robustness of our work. We genuinely appreciate your thorough review and your positive assessment of our efforts.

However, the major shortcoming still persists, due to the mixed material classifications: “nano-sized glass fiber-reinforced composite and different types of composites (SFRC composites, low/ high-viscosity bulk-fill composites and microhybrid composite).” Consistant classification criterion is key.

Response: Thanks for the reviewer for helpful feedback. In response to your concern regarding the material classifications, we have taken your suggestion to heart. We have now reclassified the composites into two distinct categories: "fiber-reinforced" and "particle-filled," providing a consistent classification criterion throughout the manuscript. This reclassification is now clearly presented in Table 1, where an additional column has been added to reflect this categorization approach.

We believe that this consistent classification enhances the clarity and cohesiveness of our paper, allowing readers to better grasp the distinctions among various composite types. Your emphasis on this matter has undoubtedly contributed to the improved quality of our work.

Once again, we sincerely thank you for your dedicated review and insightful comments.

The result presentation and discussion need to be modified, because currently the MS has been written for speciallised dentistry journals, comparing commercial products to inform clinical practice. However, to qualify for publicaiton in a multidisciplinary journal Polymers, this needs to be modifed, because in the broader dental materials and polymer science context each individual commercial name does not convey meaningful information.

Response: Thanks for the reviewer for helpful feedback. Thank you for your valuable feedback. We greatly appreciate your insights, which guided us in refining our manuscript for Polymers journal. Based on your suggestions, we have revised the result presentation and discussion sections to align better with the broader context of dental materials and polymer science.

We have adopted a more general approach by focusing on the overall characteristics and attributes of the commercial products, rather than specific brand names. This adjustment ensures that our work is more in line with the multidisciplinary nature of Polymers journal.

We believe that your feedback has significantly contributed to enhancing the quality and relevance of our paper. Thank you once again for your time and thoughtful input.

The null hypothesis has not been revisited. Also this should be changed to reflect the main findings of the study, the correlation. Perhaps the title should also be changed. Thus, a major revision is recommended.

Response: Thanks for the reviewer for helpful feedback. Your guidance for a major revision is well-received, and we are committed to addressing these aspects to enhance the quality and impact of our paper. In response to your suggestions, we will thoroughly revisit the null hypothesis.

Details: Please use consistant classificaiton, based on glass fiber types (also since the focus is primarily “on fiber type”): roughly nHAglass, glass, glass+zirconia, mixed glasses, etc. And if the authors prefer (and secondary focus is on “length parameters”) further classification could be made based on length scale: mm, um, nm, etc. Please read the literature to find appropriate classifications.

Response: Thanks for the reviewer for helpful feedback. Following your advice, we have revisited the classification of materials in our study.

Table 1 should be accordingly modified, which can definitely use more columns to increase the readability, e.g. for classification, manufacturer, filler types, filler %. Take out word“group” througout.

Response: Thanks for the reviewer for helpful feedback. As per your valuable suggestion, we have appropriately modified Table 1 to enhance readability. We have added additional columns to include classification, manufacturer, filler types, and filler percentages. Additionally, we have removed the word "group" throughout the table, in accordance with your guidance.

In Abstract the introduction of the 8 commercial samples needs to be consistent because right now some have the manufactuers, and some do not. And it is not clear which one is the “newly developed nano-sized glass fiber-reinforced composite.”

Response: Thanks for the reviewer for helpful feedback. In response to your feedback, we have revised the Abstract to ensure consistency in introducing the 8 commercial samples. We have made sure that all samples are presented in a uniform manner, with appropriate manufacturer information included. Additionally, we have clarified the identification of the "newly developed nano-sized glass fiber-reinforced composite" for better clarity.

Two styles of decimal point have been used. Please only use one. Please use proper multiply sign rather than letter x.

Response: Thanks for the reviewer for helpful feedback. We have taken your feedback into consideration and have ensured consistency in the use of decimal points throughout the manuscript. Additionally, we have replaced the letter "x" with the proper multiply sign as advised. Thank you for your precise observations.

The highly impacting work depicting the mechanical properties of restorative dental composites in general, factors influencing the properties, filler-matrix interface and filler reinforcement effect should be cited doi:10.1038/ncomms9631

Response: Thanks for the reviewer for helpful feedback. We extend our gratitude for your thorough review of our manuscript and for pointing us towards this impactful work.

However, upon further examination, we realized that the provided DOI corresponds to a study focused on glass ionomer cements rather than restorative dental composites, as our paper primarily addresses the latter.

Nonetheless, we highly value your recommendation and are committed to appropriately incorporating this valuable work into our research. In our investigations related to glass ionomer cements, we will ensure to include a reference to the mentioned paper (doi:10.1038/ncomms9631) to acknowledge its impact in the field.

We are thankful for your understanding and for the time you've dedicated to reviewing our manuscript. Your insights have significantly contributed to the quality of our work.

Round 3

Reviewer 1 Report

accept